# Evaluation of Quality Parameters and Functional Activity of Ottobratica Extra Virgin Olive Oil Enriched with *Zingiber officinale* (Ginger) by Two Different Enrichment Processes during One-Year Storage

**DOI:** 10.3390/foods12203822

**Published:** 2023-10-18

**Authors:** Irene Maria Grazia Custureri, Vincenzo Sicari, Monica Rosa Loizzo, Rosa Tundis, Ana Cristina Soria, Angelo Maria Giuffrè

**Affiliations:** 1Department of Agraria, University “Mediterranea” of Reggio Calabria, Salita Melissari, Feo di Vito, 89124 Reggio Calabria, Italy; irene.custureri@unirc.it (I.M.G.C.); amgiuffre@unirc.it (A.M.G.); 2Department of Pharmacy, Health and Nutritional Sciences, University of Calabria, Edificio Polifunzionale, Via P. Bucci, 87036 Rende, Italy; rosa.tundis@unical.it; 3Institute of General Organic Chemistry (IQOG-CSIC), Juan de la Cierva 3, 28006 Madrid, Spain; acsoria@iqog.csic.es

**Keywords:** extra virgin olive oil, functional olive oil, ginger, antioxidant activity, anti-obesity effect, sensory analysis

## Abstract

The aim of this work was to evaluate the impact of two enrichment processes on the quality parameters and bioactivity of Ottobratica extra virgin olive oil (EVOO) with ginger during storage. The first procedure was conducted by including ginger powder with olive fruits in the malaxer, and the second by infusion into the EVOO. The obtained oils were stored at room temperature for one year in the dark and periodically analysed. To evaluate the effect on the shelf-life of flavoured olive oils (FVOOs), physical, chemical and sensory parameters were evaluated. The FVOOs were investigated for antioxidant activity through a multi-target approach. The inhibition of lipase and carbohydrate hydrolysing enzymes was analysed. The addition of ginger in the malaxer generated a product that preserved the lowest values of peroxide after storage (10.57 mEq O_2_ kg^−1^) and maintained the highest α-tocopherol level (101.16 mg kg^−1^). The FVOOs, regardless of the enrichment technique used, showed a higher antioxidant activity than EVOO. Generally, a reduction in the inhibitory activity of the carbohydrate inhibitory enzymes was observed, especially after 60 days of storage. The addition of ginger improved the lipase inhibitory effect, especially if added during malaxation, and helped the FVOOs maintain this activity during storage.

## 1. Introduction

EVOO (extra virgin olive oil), one of the most important products of the Mediterranean diet, helps human health by preventing free radicals thanks to its content of unsaturated fatty acids (both monounsaturated fatty acids, MUFAs, and polyunsaturated fatty acids, PUFAs) and its phenolic compounds, which comprise only ~2% of EVOO. This health claim was approved by the European Food Safety Authority (EFSA) with Directive n. 432/2012 [1]. The development and testing of olive oils to which functional molecules have been added is very interesting, mainly considering the recent increase in food-related pathologies such as eating disorders [2]. The resulting product cannot be labelled as ‘extra virgin olive oil’, but can be labelled instead as flavoured virgin olive oil (FVOO). These FVOOs are characterized by an improved nutritional value, enriched sensory characteristics and an increased shelf-life. An analysis of the literature revealed that there is a great variability in the aromatization process of an EVOO. Some authors compared the impact of different production techniques on the quality of the derived FVOOs [3]. The results clearly showed that adding the selected extract during malaxation is not only an eco-friendly and solvent-free method, faster and easier than infusion, for example, but is also more efficient at extracting phenolic compounds, with significantly reduced levels of hydrolysis [4].

Calabria is one of the main olive oil producing regions in Italy. The climate is mild, typical of the Mediterranean area, and favourable for olive tree cultivation. It is rich in autochthonous varieties, grown in the different areas of the region. One of the most popular of these cultivars is Ottobratica, mainly present in the Tyrrhenian area of the region, whose oil has very low acidity values. This is due not only to genetic factors but also to the climate in its area of cultivation, which never reaches high temperatures or humidity [5,6]. When compared to other autochthonous Calabrian varieties, Ottobratica oil shows the highest total phenolic content, and medium to high tocopherols levels [7].

Previous studies conducted in the same geographical area and on various olive cultivars (including Ottobratica) have demonstrated that acidity and oxidative-related parameters are related to pre- and post-harvest variables [8] such as cultivar and harvest date [9]. Additionally, the biometric parameters, such as weight of fruit, pulp/seed ratio and water and oil content, are also related to cultivar and harvest date [10]. This is very important because the extractive parameters of the industrial plant (malaxation duration, pressure and pressing duration) are related to these parameters.

Ginger (*Zingiber officinale* Rosco) belongs to *Zingiberaceae* family. The rhizome is widely used as a spice for its flavour and also as a medicinal plant. Ginger is rich in bioactive phenolics, in particular gingerols and shogaols, which are responsible for its bitter taste. Different authors have demonstrated their positive effect on human health [11].

The market for enriched oils has been growing in recent years. Consumers are increasingly interested in the health properties of foods, and studies have shown that they are particularly curious about functional oils [12]. According to Hamam et al. [12], 60% of surveyed consumers would pay an extra sum of money for a vitaminized olive oil [12]. 

In recent years, obesity has increased worldwide, and it is known to be frequently associated with diabetes [13]. These conditions indicate a general metabolic disorder. It is of great importance to reduce the absorption of sugar and fat. One of the most common practices to do it is to reduce their absorption in the intestinal tract using pancreatic lipase and carbohydrates hydrolysing enzymes [14]. Foods naturally rich in molecules capable of positively affect the digestion of lipids or carbohydrates and possessing anti-obesity properties are thus highly valued [15]. Rodríguez-Pérez et al. [15] stated that in vitro tests are a good starting plan for the treatment of obesity. Moreover, they are useful for identifying which plant extracts are more active or richest in single polyphenols with this specific function. There are few in vivo studies confirming the real beneficial potential of olive oil in treating this disease [15].

Although adding plant material or a spice into an olive oil by infusion is obviously one of the easiest, quickest and most affordable methods, several authors [3,4] confirmed that adding these to the malaxer is more efficient in terms of the bioactivity of the final product. The aim of this study is to test the repeatability of ginger as an enrichment matrix in olive oil. The Ottobratica variety is of significant economic importance for the territories in which it typically grows and it has the advantage of early fruiting if compared to other varieties cultivated in the same geographical area. Moreover, it is interesting to understand how an Ottobratica olive oil, already naturally rich in polyphenols, behaves when enriched. 

Other authors have previously studied olive oil enriched with ginger [16,17], but none examined the evolution of the quality parameters and the enzymatic activity during one year of storage, focusing more on the volatile profile. Thus, this article reports the impact of technological enrichment processes on the quality parameters and bioactivity of an EVOO obtained from the autochthonous Calabrian cultivar “Ottobratica” flavoured with ginger *(Zingiber officinale* R.). The obtained FVOOs were monitored throughout 360 days of storage. For enrichment processes, two different techniques were applied: one was conducted by including ginger root powder with olive fruits in the malaxer and the other one by infusion for 30 days in the dark. The obtained FVOOs were stored at 25 °C for 360 days and periodically analysed for the evaluation of quality parameters and bioactivity in terms of antioxidants and inhibition of key enzymes linked to type 2 diabetes and obesity. 

## 2. Materials and Methods

### 2.1. Samples

Olive fruits (*Olea europea* L.) from the Ottobratica cultivar were harvested near Polistena in the province of Reggio Calabria during the 2021 crop season. The olives were randomly picked by a mechanical shaker from five trees of between 20 and 30 years of age. The fruits were placed in plastic boxes (20 kg each) and processed in the following 24 h. The oil extraction was conducted by pressure of the olive paste by a laboratory apparatus (Agrimec Valpesana, Calzaiolo, San Casciano, Florence, Italy).

The capacity of the system was around 20 kg per milling. Ginger root powder was purchased at a local supermarket, packaged by Silanpepe in little plastic bag with a capacity of 150 g, year of production 2020. It was added (1%) during olive paste malaxation, which was conducted at room temperature. The pressure was slowly increased to a maximum of 200 atm (20 min); the extraction procedure was 40 min. The oily phase was recovered, centrifuged and filtered using a paper filter. 

A concurrent EVOO (extra virgin olive oil) enrichment was also conducted by infusion (2% ginger root powder in a sterile gauze bag) for 30 days, in the dark and with constant mechanical shaking. Both FVOOs (malaxation enriched and infusion enriched) were stored at room temperature, in the dark, in 100 mL green glass bottles with a screw cap.

The physical, chemical and sensory analyses were conducted on the EVOO (control), on the sample enriched during malaxation (GM) and on the sample enriched by infusion (GI).

Analyses were conducted at the following times: T0 (day of production); T15 (15 days after production); T30 (after 30 days); T60 (after 60 days); T180 (after 180 days); and T360 (after 360 days).

### 2.2. Analytical Methods

#### 2.2.1. Ginger Powder, EVOO and FVOO Extraction Procedure

The extraction of ginger was performed by ultrasound-assisted probe technology as suggested by Contreras-López et al. [18] with some modification. Approximately 5 g of powdered ginger root was placed in a tube with 100 mL of distilled water. A 25 mm probe was introduced. The extraction was conducted for 15 min at a pulse mode of two seconds on/four seconds off and power of 15%. The extract was centrifuged at 8000× *g* for 10 min. The mixture was filtered with a Büchner funnel and kept at −4 °C until analysis. 

For EVOO and FVOO extraction, the procedure of Montedoro et al. [19] was applied. Oils were mixed with a hydroalcoholic solution (7:3 *v*/*v*), then treated with *n*-hexane. The residue was taken up with hydroalcoholic solution (1:1 *v*/*v*) and stored at −20 °C until analysis.

#### 2.2.2. Total Phenol Content and Total Carotenoid Content in Ginger Powder

The TPC was evaluated as previously described by Sepahpour et al. [20]. The results were expressed as mg gallic acid equivalents (GAE) g^−1^ of the extract.

For TCC, the methodology proposed by Silva de Rocha et al. was used [21]. Results are expressed as equivalent mg β-carotene g^−1^ DW plant material. 

#### 2.2.3. Free Acidity, Peroxide Value and Spectrophotometric Indices in EVOO and FVOOs

EVOO quality parameters were determined according to EEC Regulation [22]. 

#### 2.2.4. Total Phenol of EVOO and FVOOs

The total phenols content (TPC) was determined using Folin–Ciocalteu method [23].

#### 2.2.5. Colour in EVOO and FVOOs

The colour was measured with a colorimeter (Konica Minolta CM-700d, Osaka, Japan), according to the international standard CIE L*, a*, b*. Results were reported as chroma (C*). 

#### 2.2.6. Chlorophyll and Carotenoid in EVOO and FVOOs

Pigments were extracted from the oil samples using 5 mL of oil and 5 mL of *n*-hexane. Total contents of chlorophyll (TChlC) and carotenoid (TCC) were determined spectrophotometrically (670 nm and 470 nm, respectively) and expressed as mg kg^−1^ of oil [24].

#### 2.2.7. α-Tocopherol Content in EVOO and FVOOs

The oil samples were diluted in 2-propanol (1:10) and filtered using a syringe filter (0.45 μm pore size). An aliquot of five μL of sample was injected into an ultra-high performance liquid chromatography (UHPLC) system (UHPLC PLATINblue, Knauer, Germany) coupled with a fluorescence detector RF-20A/RF-20Axs model (Shimadzu Corporation, Kyoto, Japan) and analysed (flow rate of 0.5 mL min^−1^) through a mobile phase of methanol/acetonitrile (50:50). The detector was set at a 290 nm excitation wavelength and a 330 nm emission wavelength. The identification and quantification were performed by calibration curve, using pure α-tocopherol, and results were expressed as mg kg^−1^ of oil [25]. 

#### 2.2.8. EVOO and FVOOs Phenolic Profile

For the individual quantification of phenolic compounds by UHPLC, two μL of antioxidant extract was injected in the UHPLC–DAD system, equipped with a binary pump system, with column C18A (1.8 μm, 100 mm × 2 mm) thermo-regulated at 30 °C during the analysis, coupled with a PDA-1 (photodiode array detector, PLATINblue); the mobile phases were water acidified with acetic acid (pH 3.1) and acetonitrile, and the flow rate correspond to 0.4 mL min^−1^. The detector was set at a 254, 280, 330, 350 and 450 nm wavelengths. For the quantification, external standards purchased from Merck (Darmstadt, Germany) were used and the results were expressed as mg kg^−1^ [26]. 

#### 2.2.9. ABTS and DPPH Tests

The ABTS test was applied to investigate the radical scavenging ability of the samples using a procedure previously described [27]. The absorbance was measured at 734 nm. 

The DPPH radical scavenging assay was applied using the procedure previously described [27]. Ascorbic acid was used as the positive control in both radical scavenging assays.

#### 2.2.10. β-Carotene Bleaching Test

The β-carotene bleaching test was done following the procedure previously described [28]. The absorbance was read at λ = 470 nm. 

#### 2.2.11. FRAP

For antioxidant determination through FRAP assay, the method described by Plastina et al. [28] was adopted. The absorbance was measured at 595 nm. 

#### 2.2.12. Carbohydrate Hydrolysing Enzyme Inhibitory Effect

The α-amylase inhibitory activity of PSPs was determined using the method of Tundis et al. [29]. The absorbance was read at 500 nm.

#### 2.2.13. Pancreatic Lipase Inhibitory Effect

Pancreatic lipase inhibitory activity was determined as previously described using orlistat as a positive control [28].

#### 2.2.14. Sensory Analysis

The panel was made up of seven specialist assessors (age range: 30 to 65 years). The evaluation was carried out using a 9-point structured scale where 1 is absent and 9 is extremely perceptible. The quantitative method (QDA) was performed to define the sensory profile of each sample. QDA test results were analysed and reported in a graphical spider plot using Microsoft Office Excel 2014.

### 2.3. Statistical Analysis

Samples were analysed in triplicate. Analytical data were reported as means ± standard deviation. The analysis of variance (one-way ANOVA) was conducted by applying the post hoc Tukey test at *p* < 0.01 (SPSS software, 21.0 version, Armonk, NY, USA). The following symbols were used to indicate the significance: * *p* ≤ 0.05; ** *p* ≤ 0.01; *p* > 0.05; ns, not significant.

## 3. Results and Discussion

### 3.1. Free Acidity, Peroxide Value and Spectrophotometric Indices

The free acidity values (Figure 1) showed how the EVOO (extra virgin olive oil) fell within the values stipulated for extra virgin olive oils (≤0.80%) [22]. The variations over time were highly significant (*p* < 0.01) and increased from 0.68% at T0 to 0.84% at T360. The addition of ginger caused an increase in FA (free acidity) in both the FVOOs (flavoured virgin olive oils). Concerning this value, contradictory data are present in the literature: the value depends on the spices, on the cultivar of the olive oil enriched and on the procedure employed [30]. For example, Ayadi et al. [31] supplemented a Tunisian extra virgin olive with several aromatic plants, and noted an increase in FA in all the mixtures; these data are in accordance with our results [31]. Likewise, Sousa et al. [32] noticed that the addition of garlic also caused an increase in FA in the flavoured sample [32].

Oil enriched by infusion (GI) showed the same FA values as EVOO only at the 15-day storage check; after that, the FA increased rapidly and the values of the GI were always higher than the EVOO (*p* < 0.01). Oil enriched during malaxation (GM) always showed significantly higher values than EVOO (*p* < 0.01), between 0.84% (T0) and 1.4% (T360) and almost always higher than GI. Values of GM increased by 166% in one-year storage and 219% when compared with EVOO at T0.

Peroxide values (PV) are described in Figure 2. The values for EVOO, during one year of storage, increased from 9.45 to 17.86 mEq O_2_ kg^−1^ (*p* < 0.01), i.e., within the maximum value stated by the European Union Commission (2016) for an EVOO (20 mEq O_2_ kg^−1^) and equivalent to findings of other authors for oil of the Ottobratica cultivar [22,33,34]. GI always showed values slightly lower than EVOO, whereas GM showed the best performance in this regard, increasing from 6.92 (T0) to 10.57 mEq O_2_ kg^−1^ (T360) (*p* < 0.01). At each sampling, the differences between PV were significantly different (*p* < 0.01), showing a significant influence of variables.

Concerning the value of conjugated diene and triene, K_232_ and K_268_ (Figure 3 and Figure 4), the results are partially in accordance with the results of Moustakime et al. (2021) who showed how different aromatization techniques lead to a decrease in the content of diene and an increase in the content of triene conjugated [35]. In our study, in the first 30 days of storage, both GI and GM had a level of K_232_ lower than the control. After 360 days, both extinction coefficients were significantly higher in GM and GI. K_268_ in GM started to notably increase from the sixth month, which is in accordance with previously reported data [36]. A similar trend in the sample was observed regarding ΔK during storage (Table 1).

### 3.2. Colour, Chlorophyll and Carotenoid

A fundamental parameter for consumer acceptability is the colour. In the EVOO and FVOOs there was a significant decrease in chroma C*, about four times lower than T0 (Figure 5). The addition of spices can increase chlorophyll and carotenoid content of the flavoured oils. However, there is a consequent change to the colour of the oil, and therefore the acceptability to the consumer [30,37]. Chlorophyll gives a greenish colouration and carotenoid compounds are responsible for a yellowish coloration. The content of chlorophyll and carotenoid in an oil is highly variable. It varies according to the cultivar, to the level of ripeness of the olives, the extraction technique used and the methods of conservation of the oil. Pigments in olive oil are directly related to oxidative stability [38]. Tuberoso et al. [39] found a great variability between Sardinian cultivars in chlorophyll ranging from 6.5 for Semidana to a maximum of 10.8 mg kg^−1^ of oil for Bosana. The same is true for carotenoid, for which the same authors found levels ranging from 20.9 for the Semidana cv to a maximum of 47.6 mg kg^−1^ of oil for the Tonda di Cagliari cv [39]. Figure 6 and Figure 7 show the pigment content in the control and the FVOOs during storage. The evaluation of total chlorophyll content (TChlC) and total carotenoid content (TCC) showed high values in the first 30 days of storage and a natural decrease after 60 days of storage, reaching values after 360 days of storage for TChlC of 11.03 and TCC of 4.80 mg kg^−1^ of oil, TChlC of 11.20 and TCC of 4.96 mg kg^−1^ of oil and TChlC of 14.10 and TCC of 6.33 mg kg^−1^ of oil in EVOO, GM and GI, respectively. In general, TChlC was more influenced than TCC compared to the unflavoured oil. 

### 3.3. TPC, α-Tocopherol Content and Individual Phenols by UHPLC

Total phenolic content (TPC) (Figure 8) of EVOO corresponded to 418.51 mg gallic acid (GAE) kg^−1^ of oil, lower than that found by De Bruno et al. (1150 mg GAE kg^−1^) [25], but in accordance with the quantity found by Piscopo et al. (469 mg GAE kg^−1^) [40], who explained that the TPC varies also with the storage temperatures of the olives. As expected, the enrichment of EVOO improves its quantity of phenols, especially when the matrix was added in the olive paste. However, this type of addition might cause an increment in paste volume and a naturally greater loss in the olive mill wastewater [41]. In fact, GM sample showed a lower phenolic content than the unflavoured sample at T0. The TPC data analysis showed that during storage the following trend should be observed: GI > EVOO > GM. The difference in TPC in GI and GM showed that the infusion procedure seems to be better than addition during malaxation to enrich oils with these phytochemicals. Table 2 shows the bioactivity of ginger extract.

α-Tocopherol is a vitamin with antioxidant properties and plays an important role against cellular autoxidation and oxygen radicals. It is sensitive to heat and light and it degrades in the presence of high temperatures. EVOO is naturally rich in tocopherols. The literature records high levels of α-tocopherol in Calabrian autochthonous cultivars, such as Grossa di Gerace, which may reach a value of 365 ppm, and Ottobratica, which may reach a value of 330 ppm [40]. Other authors reported that the abundant natural active substances in the addition matrix act synergistically as scavengers of free radicals and contribute to the protection against degradation by thermal oxidation [35]. These natural components (depending on the plant material) can react with free radicals in olive oil, thus effectively inhibiting the loss of tocopherols. The trends of α-tocopherol content in all samples during storage are reported in Table 3. The initial level of α-tocopherol for the control, which was in accordance with the literature, corresponded to 354.63 mg kg^−1^ and the lowest value was observed for GM (317.81 mg kg^−1^). Starting from T180, a large decrease was observed, and GM again showed the lowest value. After one year of storage, the α-tocopherol content decreased significantly, reaching values of 79.53, 101.96 and 85.48 mg kg^−1^ for EVOO, GM and GI, respectively. Although GM had the lowest values during the totality of storage, at the end it demonstrated the best protective effect against the loss of this molecule, even though both FVOOs maintained a higher level than the control. However, the combination of the ginger with the olive paste provides the most promising data, with a similar trend found in the literature for enrichment with goji berries [42]. 

UHPLC analysis provided identification of individual phenols of giner extract, unflavoured and flavoured oils. Table 4 and Appendix A show the single phenolic composition of ginger extract. EVOO was characterized by a high amount of pinoresinol (43.38 mg kg^−1^), hydroxytyrosol (16.15 mg kg^−1^) and tyrosol (15.61 mg kg^−1^), and a low quantity of oleoropein (0.86 mg kg^−1^) (Table 5 and Appendix A). It is known that during one year of storage, a single phenol of an olive oil may undergo an increase or decrease caused by complex hydrolytic or enzymatic activities [43]. Data analysis shows that hydroxytyrosol and tyrosol content almost doubled, whereas oleuropein content showed a fourfold decrease, and pinoresinol content remained constant. The rest of the phenols followed an opposite trend with a reduction during storage. Regarding the FVOOs (Table 6 and Table 7 and Appendix A), 6-gingerol and 6-shogaol were present in different concentrations according to whether enrichment was carried out by infusion or during malaxation. Among the main compounds from the matrix, the highest content was represented by 6-gingerol, as well as in the ginger extract. In GM, the typical trend of hydroxytyrosol and tyrosol was not observed and there was an exponential increment of 6-gingerol during the 12 months of storage. For 6-shogaol, different studies have confirmed the in vivo and in vitro activity against lipid absorption [44]. However, for GI, the content of phenols from ginger remained constant throughout storage.

### 3.4. Antioxidant Activity

Table 8 and Table 9, and Figure 9, show EVOO and FVOOs antioxidant activity. In general, EVOO showed a good radical scavenging activity, although this decreased during storage, with IC_50_ values for the DPPH assay from 12.33 to 29.54 μg mL^−1^ at T0 and T360, respectively. Values from 3.43 to 15.21 μg mL^−1^ at T0 and T360, respectively, were found with the ABTS test. A great variability in antioxidant activity was observed in the extract derived from EVOO obtained from the Frantoio cultivar [24]. The values for this cultivar ranged from 45.3 to 256.8 and from 56.3 to 279.6 μg mL^−1^ for DPPH and ABTS, respectively. FVOOs obtained from both procedures were richer in phytochemicals and able to counteract DPPH and ABTS radicals, especially after 360 days of storage, with IC_50_ values of 44.21 and 35.35 μg mL^−1^ for GI and GM, respectively, for the DPPH test. A similar situation was also observed in the ABTS test, with IC_50_ values of 11.31 and 26.31 μg mL^−1^ for GI and GM, respectively. It is interesting to note that ginger protected oil from losing the ability to protect from lipid peroxidation. In fact, the IC_50_ value in EVOO passes from 48.72 to >100 μg mL^−1^ at T0 and T360, respectively, whereas values were 18.68–46.10 μg mL^−1^ at T0 and 18.68–77.67 μg mL^−1^ at T360 for GI and GM, respectively. FRAP assay data show that, regardless of the storage time, the results are lower than the BHT positive control 63.26 μM Fe(II) g^−1^ for the EVOO (from 25.01 to 4.31 μM Fe(II) g^−1^ at T0 and T360, respectively). GM was the most active sample in terms of iron reduction power regardless of the storage time. Previously, Loizzo et al. [45] reported the radical scavenging potential of FVOO obtained by adding Capsicum chinense and *C. annuum* fine dry powder to Carolea extra virgin olive oil by infusion. FVOO formulated with Aji limo was the most active, with IC_50_ values of 18.8 and 27.6 μg mL^−1^ in DPPH and ABTS test, respectively. Moreover, the addition of red peppers significantly improved FRAP activity, with FRAP values ranging from 129.8–139.5 μM Fe(II) g^−1^ for FVOO with Red Topepo and Red mushroom, respectively. 

### 3.5. Inhibitory Activity against Key Enzymes Linked to Type 2 Diabetes and Obesity

Samples were also tested to evaluate the potential inhibitory activity against carbohydrate hydrolysing enzymes α-amylase and α-glucosidase (Table 10). In the unflavoured samples, the IC_50_ values obtained in the α-amylase and α-glucosidase tests were compared to the positive control, with values for α-amylase from 269.02 to 289.32 μg mL^−1^ at T0 and T360, respectively, and for α-glucosidase from 137.34 to 778.23 μg mL^−1^, respectively.

Generally, independently of the technological processes used for the enrichment and the enzyme used, a reduction in the inhibitory activity was observed, especially at T60. However, it is interesting to note that FVOOs are characterized by a higher inhibitory activity than EVOO, with IC_50_ values of 205.11 and 228.10 μg mL^−1^ for GI and GM, respectively, compared to 389.32 μg mL^−1^ against α-amylase. A similar observation can be made for α-glucosidase. In general, the addition of ginger does not improve the potency of the oil’s activity on the enzymes responsible for the breakdown of carbohydrates but it does help maintain its functional properties even after 360 days of storage.

The hypolipidemic activity (Table 11) was evaluated by the inhibition of pancreatic lipase. This enzyme intervenes in the metabolism of fats and its inhibition determines a better control of the lipid profile. From the analysis of the results, it is possible to see that the addition of ginger powder extract improves the enzyme inhibitory effect with IC_50_ values of 63.45 and 54.48 μg mL^−1^ for GI and GM, respectively, compared to 143.46 μg mL^−1^ for EVOO. Its addition during malaxation resulted in a better product in terms of hypolipidemic effect since the IC_50_ values always remained lower than GI for the same sampling period and reached a value of 119.21 μg mL^−1^ by the end of observation (T360).

### 3.6. Sensory Analysis

EVOOs and FVOOs were characterized by sensory analysis. This is an interesting test because consumers often continue to prefer a traditional unflavoured oil, even when a new, flavoured product is created. The panel members identified the oils enriched with ginger during malaxation but not the FVOOs produced by infusion. First of all, the FVOOs scored an overall acceptability of 6 and 7 points for GI and GM, respectively. All the defects, in particularly the “rancidity” of the control, were well covered. This result means that ginger volatiles have a masking effect on olives with slight off-flavours. Figure 10 and Figure 11 report the sensory profile of EVOO and FVOOs, and show different changes in olfactory and gustatory sensations. In the FVOOs, new sensory descriptors were added like “pungent”, “smoked”, “citrusy”, “astringent” and “spicy”. Regarding the olfactory sensations, the most evident differences were in the “green fruity” descriptor, which decreased as a consequence of flavouring [46]. “Pungent” and “smoked” appeared particularly in GI, whereas “citrusy” and “vegetable note” appeared in GM. The characteristic “green fruity” typical of the Ottobratica olive oil cultivar was partially lost. In the gustatory sensation, the taste typical of ginger greatly increased, in particular for GM. Also interesting is the increment of the attributes “sweet” and “floral”, which significantly increased compared to the unflavoured sample. GM was also positively evaluated for its general equilibrium in all the new notes. Hamam et al. [12] claimed that about 60% of consumers would pay more for an enriched olive oil and that the sensory attributes play a key role in their purchasing decisions. 

## 4. Conclusions

In the development of these types of products, which may be considered a kind of food supplement, it is of primary importance to adopt a multi-analytic plan to provide a more complete characterization and find the best formulation with the highest bioactivity. The obtained results demonstrate how the techniques used lead to two different products with different properties. The addition of the ginger powder directly to the olive paste rather than by infusion gives a superior flavoured product. The enrichment influenced the chemical and sensory characteristics of the new formulation, more noticeably in the case of GM (ginger flavoured olive oil by malaxation) compared to GI (ginger flavoured olive oil by infusion). The infusion sample suffered from greater oxidation during storage than the control and GM. Regarding the protection of lipid peroxidation, GM and GI both had greater activity during storage. The inhibition of the enzymatic activity showed how GM has good in vitro activity against obesity, probably due to the high content of 6-gingerol and 6-shogaol. The content of 6-shogaol increased until T60, coinciding with the best activity against pancreatic lipase, confirming its activity against lipid absorption.

## Figures and Tables

**Figure 1 foods-12-03822-f001:**
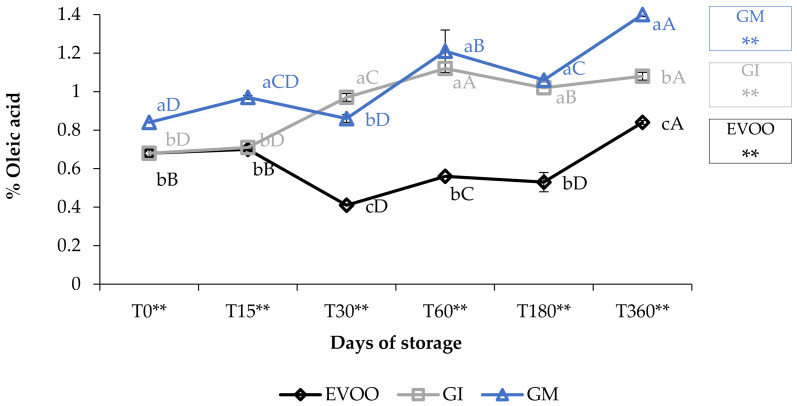
Free acidity during storage. Values are expressed as % of oleic acid. Data are expressed as means ± S.D. (*n* = 3). EVOO: control; GI: ginger olive oil obtained by infusion; GM: ginger olive oil obtained by malaxation. Results followed by different capital letters show the differences in one sample during storage. The different lowercase letters show the differences among the samples at the same time. Differences within and between groups were evaluated by one-way ANOVA followed by Tukey’s test: ** *p* < 0.01. Results followed by different letters are highly significantly different at * *p* ≤ 0.05; ** *p* ≤ 0.01; ns *p* > 0.05 not significant.

**Figure 2 foods-12-03822-f002:**
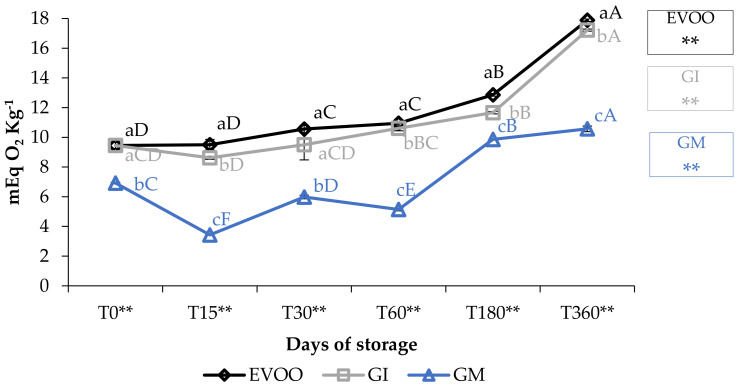
Peroxide values during storage. Values are expressed as mEq O_2_ kg^−1^. Data are expressed as means ± S.D. (*n* = 3). EVOO: control; GI: ginger olive oil obtained by infusion; GM: ginger olive oil obtained by malaxation. Results followed by different capital letters show the differences in one sample during storage. The different lowercase letters show the differences among the samples at the same time. Differences within and between groups were evaluated by one-way ANOVA followed by Tukey’s test: ** *p* < 0.01. Results followed by different letters are highly significantly different at * *p* ≤ 0.05; ** *p* ≤ 0.01; ns *p* > 0.05 not significant.

**Figure 3 foods-12-03822-f003:**
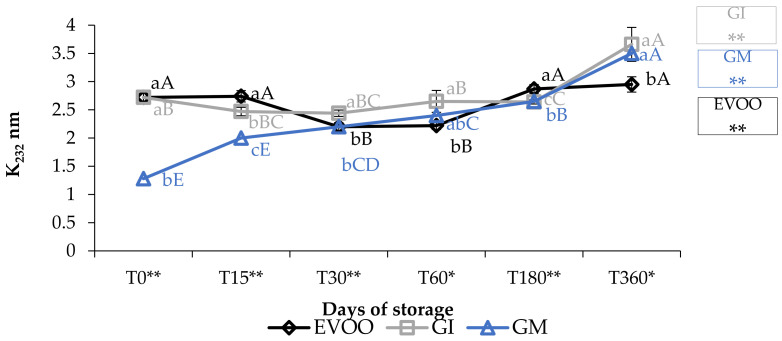
K_232_ during storage. Data are expressed as means ± S.D. (*n* = 3). EVOO: control; GI: ginger olive oil obtained by infusion; GM: ginger olive oil obtained by malaxation. Results followed by different capital letters show the differences in one sample during storage. The different lowercase letters show the differences among the samples at the same time. Differences within and between groups were evaluated by one-way ANOVA followed by Tukey’s test: ** *p* < 0.01. Results followed by different letters are highly significantly different at * *p* ≤ 0.05; ** *p* ≤ 0.01; ns *p* > 0.05 not significant.

**Figure 4 foods-12-03822-f004:**
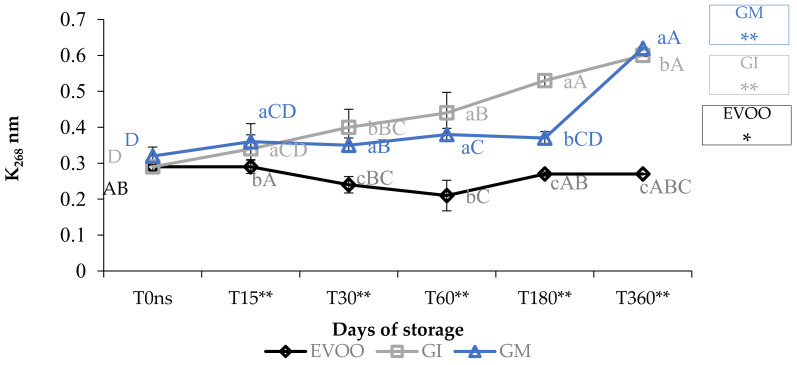
K_268_ during storage. Data are expressed as means ± S.D. (*n* = 3). EVOO: control; GI: ginger olive oil obtained by infusion; GM: ginger olive oil obtained by malaxation. Results followed by different capital letters show the differences in one sample during storage. The different lowercase letters show the differences among the samples at the same time. Differences within and between groups were evaluated by one-way ANOVA followed by Tukey’s test: ** *p* < 0.01. Results followed by different letters are highly significantly different at * *p* ≤ 0.05; ** *p* ≤ 0.01; ns *p* > 0.05 not significant.

**Figure 5 foods-12-03822-f005:**
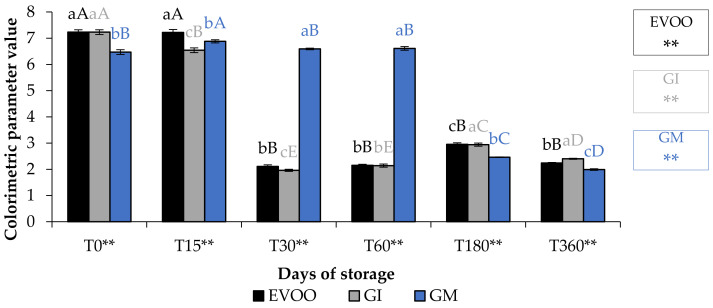
Chroma* during storage. Data are expressed as means ± S.D. (*n* = 3). EVOO: control; GI: ginger olive oil obtained by infusion; GM: ginger olive oil obtained by malaxation. Results followed by different capital letters show the differences in one sample during storage. The different lowercase letters show the differences among the samples at the same time. Differences within and between groups were evaluated by one-way ANOVA followed by Tukey’s test: ** *p* < 0.01. Results followed by different letters are highly significantly different at * *p* ≤ 0.05; ** *p* ≤ 0.01; ns *p* > 0.05 not significant.

**Figure 6 foods-12-03822-f006:**
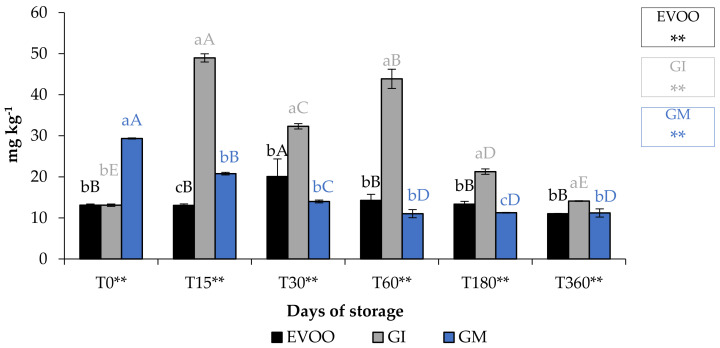
TChlC during storage. Values are expressed as mg kg^−1^. Data are expressed as means ± S.D. (*n* = 3). EVOO: control; GI: ginger olive oil obtained by infusion; GM: ginger olive oil obtained by malaxation. Results followed by different capital letters show the differences in one sample during storage. The different lowercase letters show the differences among the samples at the same time. Differences within and between groups were evaluated by one-way ANOVA followed by Tukey’s test: ** *p* < 0.01. Results followed by different letters are highly significantly different at * *p* ≤ 0.05; ** *p* ≤ 0.01; ns *p* > 0.05 not significant.

**Figure 7 foods-12-03822-f007:**
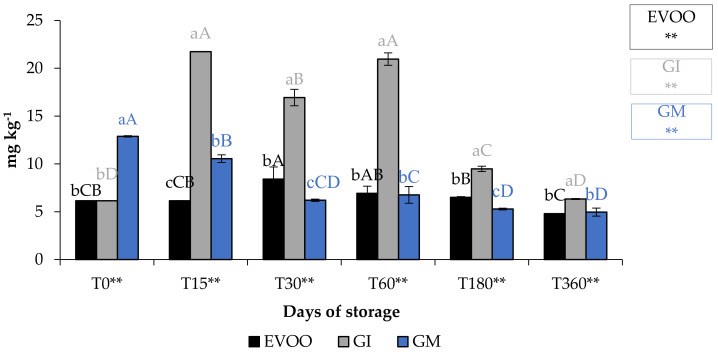
TCC during storage. Values are expressed as mg kg^−1^. Data are expressed as means ± S.D. (*n* = 3). EVOO: control; GI: ginger olive oil obtained by infusion; GM: ginger olive oil obtained by malaxation. Results followed by different capital letters show the differences in one sample during storage. The different lowercase letters show the differences among the samples at the same time. Differences within and between groups were evaluated by one-way ANOVA followed by Tukey’s test: ** *p* < 0.01. Results followed by different letters are highly significantly different at * *p* ≤ 0.05; ** *p* ≤ 0.01; ns *p* > 0.05 not significant.

**Figure 8 foods-12-03822-f008:**
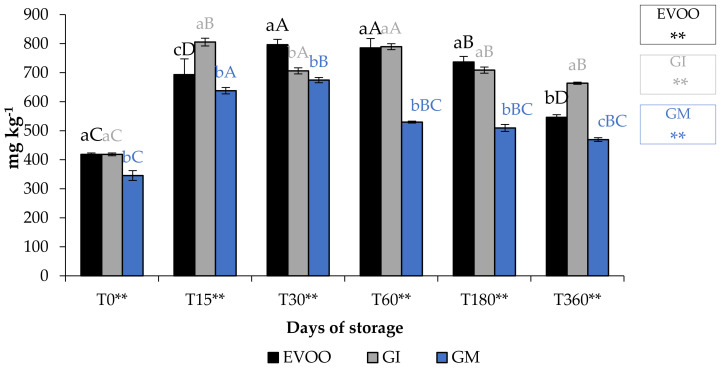
EVOO and FVOOs TPC during storage. Values are expressed as mg GAE kg^−1^. Values are expressed as mg kg^−1^. Data are expressed as means ± S.D. (*n* = 3). EVOO: control; GI: ginger olive oil obtained by infusion; GM: ginger olive oil obtained by malaxation. Results followed by different capital letters show the differences in one sample during storage. The different lowercase letters show the differences among the samples at the same time. Differences within and between groups were evaluated by one-way ANOVA followed by Tukey’s test: ** *p* < 0.01. Results followed by different letters are highly significantly different at * *p* ≤ 0.05; ** *p* ≤ 0.01; ns *p* > 0.05 not significant.

**Figure 9 foods-12-03822-f009:**
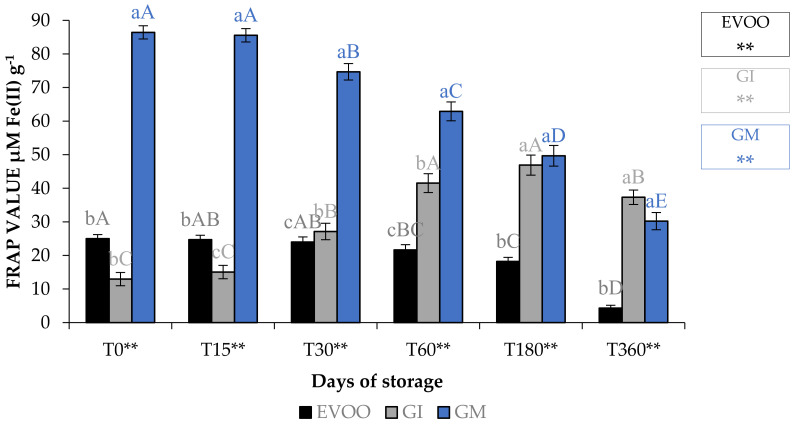
FRAP assay during storage. Values are expressed as IC_50_ (μM Fe(II) g^−1^). Data are expressed as means ± S.D. (*n* = 3). EVOO: control; GI: ginger olive oil obtained by infusion; GM: ginger olive oil obtained by malaxation. BHT (value of 63.26 ± 0.81 μM Fe(II) g^−1^) was used as positive control. Results followed by different capital letters show the differences in one sample during storage. The different lowercase letters show the differences among the samples at the same time. Differences within and between groups were evaluated by one-way ANOVA followed by Tukey’s test: ** *p* < 0.01. Results followed by different letters are highly significantly different at * *p* ≤ 0.05; ** *p* ≤ 0.01; ns *p* > 0.05 not significant.

**Figure 10 foods-12-03822-f010:**
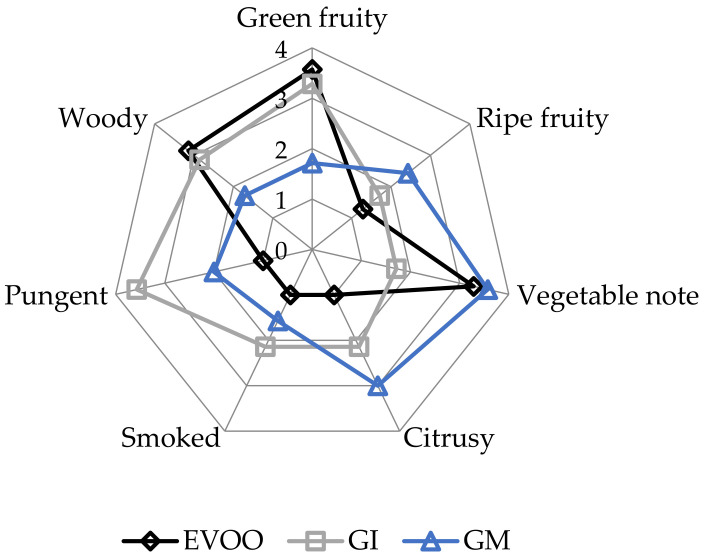
Olfactory sensations of the EVOO and FVOOs. EVOO: control; GI: ginger olive oil obtained by infusion; GM: ginger olive oil obtained by malaxation.

**Figure 11 foods-12-03822-f011:**
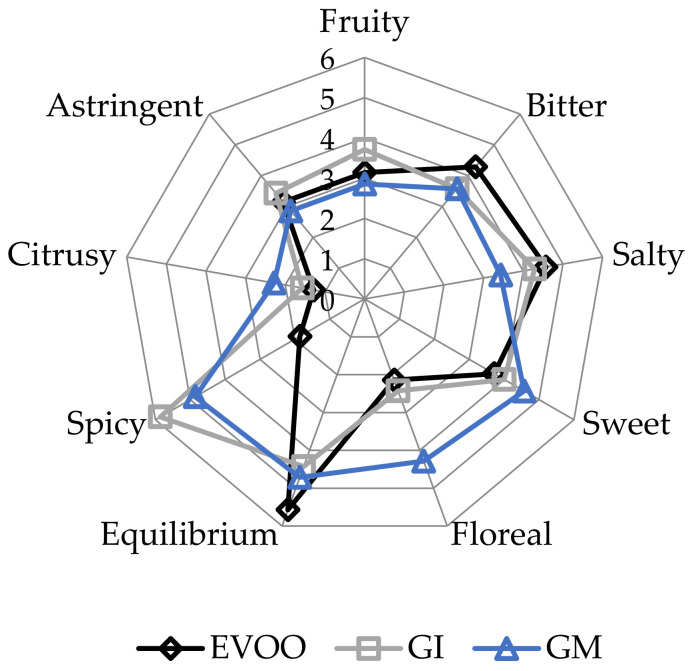
Gustatory sensations of the EVOO and FVOOs. EVOO: control; GI: ginger olive oil obtained by infusion; GM: ginger olive oil obtained by malaxation.

**Table 1 foods-12-03822-t001:** ΔK during storage. Data are expressed as means ± S.D. (*n* = 3).

	T0	T15	T30	T60	T180	T360	Sign
EVOO	−0.003 ± 0.00 ^bBC^	−0.003 ± 0.00 ^C^	−0.003 ± 0.00 ^bBC^	−0.003 ± 0.00 ^BC^	−0.001 ± 0.00 ^AB^	0.000 ± 0.00 ^bA^	**
GI	−0.003 ± 0.00 ^bAB^	−0.004 ± 0.00 ^B^	−0.004 ± 0.00 ^bB^	−0.003 ± 0.00 ^AB^	−0.001 ± 0.00 ^A^	−0.004 ± 0.00 ^cAB^	*
GM	0.001 ± 0.00 ^a^	−0.003 ± 0.00	0.000 ± 0.00 ^a^	0.003 ± 0.00	0.000 ± 0.00	0.006 ± 0.00 ^a^	ns
Sign	*	ns	**	ns	ns	**	

EVOO: control; GI: ginger olive oil obtained by infusion; GM: ginger olive oil obtained by malaxation. Results followed by different capital letters in the same row show the differences in one sample during storage. The different lowercase letters in the same column show the differences among the samples at the same time. Differences within and between groups were evaluated by one-way ANOVA followed by Tukey’s test: ** *p* < 0.01. Results followed by different letters are highly significantly different at * *p* ≤ 0.05; ** *p* ≤ 0.01; ns *p* > 0.05 not significant.

**Table 2 foods-12-03822-t002:** Bioactivity of ginger extract. Values are expressed as ^1^: mg (GAE) g^−1^; ^2^: mg β–carotene g^−1^; ^3^: IC_50_ (μg mL^−1^); ^4^: IC_50_ (μM Fe(II) g^−1^).

TPC ^1^	TCC ^2^	DPPH ^3^	ABTS ^3^	β-carotene ^3^	FRAP ^4^	α-Amylase ^3^	α-Glucosidase ^3^	Lipase ^3^
15.03 ± 1.23	19.33 ± 0.77	32.15 ± 2.15	5.32 ± 0.21	19.61 ± 2.79	46.16 ± 3.82	62.21 ± 3.26	71.46 ± 3.82	115.27 ± 4.76

Data are expressed as means ± S.D. (*n* = 3). Ascorbic acid was used as positive control in the DPPH and ABTS tests (IC_50_ values of 5.03 ± 0.82 and 1.78 ± 0.07 μg mL^−1^, respectively). Propyl gallate was used as positive control in the β–carotene bleaching test (IC_50_ values of 1.02 ± 0.01 μg mL^−1^). BHT was used as positive control in the FRAP test (IC_50_ value of 63.26 ± 0.81 μM Fe(II) g^−1^). Acarbose was used as positive control in the α-amylase and in the α-glucosidase assays (IC_50_ values of 50.18 ± 1.32 and 35.57 ± 0.99 μg mL^−1^, respectively). Orlistat was used as positive control in the lipase assay (IC_50_ value of 37.44 ± 1.08 μg mL^−1^).

**Table 3 foods-12-03822-t003:** α-Tocopherol content during storage. Values are expressed as mg kg^−1^.

	T0	T15	T30	T60	T180	T360	Sign
EVOO	354.63 ± 19.36 ^aA^	261.63 ± 45.96 ^B^	234.22 ± 64.72 ^aB^	223.72 ± 38.15 ^B^	246.61 ± 25.72 ^B^	79.53 ± 1.41 ^bC^	**
GI	351.20 ± 15.01 ^aA^	286.68 ± 23.61 ^B^	224.11 ± 23.01 ^bC^	222.70 ± 27.14 ^BC^	240.91 ± 9.26 ^C^	85.48 ± 1.06 ^abD^	**
GM	317.81 ± 9.52 ^bA^	304.49 ± 29.23 ^AB^	278.49 ± 6.80 ^aA^	271.58 ± 2.98 ^B^	210.62 ± 24.05 ^C^	101.16 ± 3.24 ^aD^	**
Sign	*	ns	**	ns	ns	**	

Data are expressed as means ± S.D. (*n* = 3). EVOO: control; GI: ginger olive oil obtained by infusion; GM: ginger olive oil obtained by malaxation. Results followed by different capital letters in the same row show the differences in one sample during storage. The different lowercase letters in the same column show the differences among the samples at the same time. Differences within and between groups were evaluated by one-way ANOVA followed by Tukey’s test: ** *p* < 0.01. Results followed by different letters are highly significantly different at * *p* ≤ 0.05; ** *p* ≤ 0.01; ns *p* > 0.05 not significant.

**Table 4 foods-12-03822-t004:** UHPLC profile of the ginger extract. Values are expressed as mg kg^−1^.

Compounds	Amount
3,4-Dihydroxybenzoic acid	192.41 ± 2.23
Vanillic acid	4.39 ± 0.23
Caffeic acid	14.13 ± 1.56
*p*-Coumaric acid	7.68 ± 1.20
Ferulic acid	8.40 ± 0.87
Rutin	106.65 ± 2.65
Quercetin	69.42 ± 1.08
Apigenin	272.70 ± 4.03
Naringenin	26.95 ± 0.95
Kaempferol	59.28 ± 0,66
Isoramnetin	96.16 ± 1.32
6-Gingerol	2058.43 ± 4.65
6-Shogaol	5.28 ± 1.11
Apigenin 7-*O*-Glucoside	263.22 ± 0.99
Gallic acid	40.12 ± 2.43
Chlorogenic acid	24.65 ± 2.09
Syringic acid	4.50 ± 0.16
Luteolin 7-*O*-Glucoside	45.06 ± 1.43

Data are expressed as means ± S.D. (*n* = 3).

**Table 5 foods-12-03822-t005:** Single phenolic compound in EVOO (control) by UHPLC. Values are expressed as mg kg^−1^.

Compounds	T0	T15	T30	T60	T180	T360	Sign
Hydroxytyrosol	16.15 ± 1.54 ^cd^	15.28 ± 0.27 ^cd^	15.17 ± 0.50 ^d^	19.46 ± 0.01 ^bc^	27.08 ± 0.95 ^a^	25.01 ± 2.50 ^ab^	**
Tyrosol	15.61 ± 2.03 ^bc^	15.11 ± 0.30 ^bc^	14.39 ± 0.93 ^bc^	18.19 ± 0.17 ^ab^	11.58 ± 1.51 ^c^	21.09 ± 0.93 ^a^	**
Vanillic acid	1.47 ± 0.02 ^a^	0.39 ± 0.09 ^d^	1.24 ± 0.08 ^c^	1.38 ± 0.12 ^b^	0.00 ^e^	0.00 ^e^	**
Homovanillic acid	1.92 ± 0.03 ^e^	2.03 ± 0.05 ^d^	3.57 ± 1.03 ^a^	2.44 ± 0.14 ^b^	2.35 ± 0.16 ^c^	1.94 ± 0.14 ^de^	**
Chlorogenic acid	1.92 ± 0.19 ^a^	1.85 ± 0.06 ^b^	1.83 ± 0.21 ^ab^	1.71 ± 0.25 ^ab^	1.65 ± 0.10 ^c^	1.60 ± 0.17 ^ab^	**
Quercetin 3,4′-Diglucoside	0.91 ± 0.07 ^b^	1.39 ± 0.16 ^a^	1.20 ± 0.16 ^ab^	1.05 ± 0.07 ^ab^	0.00 ^c^	0.00 ^c^	**
*p*-Coumaric acid	3.45 ± 0.65 ^a^	3.34 ± 0.51 ^a^	2.89 ± 0.04 ^b^	1.14 ± 0.01 ^e^	1.44 ± 0.17 ^d^	1.65 ± 0.20 ^c^	**
Luteolin-7-*O*-Glucoside	3.07 ± 0.91 ^cd^	2.41 ± 0.06 ^e^	7.42 ± 0.10 ^a^	3.39 ± 0.01 ^b^	3.05 ± 0.59 ^c^	2.99 ± 0.03 ^d^	**
Cinnamin acid	0.91 ± 0.36 ^c^	0.98 ± 0.14 ^bc^	2.73 ± 1.07 ^a^	1.08 ± 0.26 ^b^	0.54 ± 0.09 ^d^	0.61 ± 0.02 ^d^	**
Oleuropein	0.48 ± 0.08 ^b^	0.48 ± 0.05 ^b^	0.86 ± 0.37 ^a^	0.46 ± 0.02 ^b^	0.43 ± 0.06 ^b^	0.10 ± 0.01 ^c^	**
Pinoresinol	43.38 ± 0.36 ^b^	42.11 ± 3.86 ^b^	55.75 ± 3.46 ^a^	44.58 ± 1.76 ^b^	41.67 ± 1.87 ^b^	44.07 ± 1.10 ^ab^	**
Quercetin	12.94 ± 0.55 ^c^	13.00 ± 1.14 ^c^	17.17 ± 5.06 ^a^	12.26 ± 0.92 ^c^	14.73 ± 0.62 ^b^	12.93 ± 4.48 ^c^	**
Apigenin	58.98 ± 11.81 ^a^	50.64 ± 3.58 ^cb^	55.35 ± 5.42 ^b^	53.41 ± 1.94 ^bc^	49.53 ± 0.55 ^b^	53.21 ± 0.98 ^d^	**
Isoramnetin 3-*O*-Glucoside	0.12 ± 0.02 ^bc^	0.12 ± 0.03 ^c^	0.31 ± 0.15 ^a^	0.14 ± 0.02 ^b^	0.00 ^d^	0.00 ^d^	**
Apigenin 7-*O*-Glucoside	1.80 ± 0.30 ^b^	1.78 ± 0.12 ^b^	4.20 ± 2.06 ^a^	1.29 ± 0.19 ^c^	0.77 ± 0.11 ^d^	0.66 ± 0.09 ^d^	**

Data are expressed as means ± S.D. (*n* = 3). Differences within and between groups were evaluated by one-way ANOVA followed by Tukey’s test: ** *p* < 0.01. Results followed by different letters in a same line are highly significantly different at * *p* ≤ 0.05; ** *p* ≤ 0.01; ns *p* > 0.05 not significant.

**Table 6 foods-12-03822-t006:** Single phenolic compound in GI (ginger olive oil obtained by infusion) by UHPLC. Values are expressed as mg kg^−1^.

Compounds	T0	T15	T30	T60	T180	T360	Sign
Hydroxytyrosol	10.44 ± 0.44 ^b^	7.23 ± 0.11 ^c^	5.98 ± 0.20 ^c^	7.32 ± 1.50 ^c^	8.84 ± 0.54 ^c^	14.42 ± 0.80 ^a^	**
Tyrosol	9.45 ± 0.11 ^c^	12.65 ± 0.20 ^b^	11.04 ± 0.63 ^b^	11.46 ± 0.66 ^b^	9.24 ± 0.24 ^c^	18.67 ± 1.48 ^a^	**
4-Hydroxyphenyl acetate	1.08 ± 0.08 ^a^	0.94 ± 0.01 ^b^	0.53 ± 0.09 ^c^	0.82 ± 0.06 ^b^	0.00 ^c^	0.00 ^c^	**
Caffeic acid	2.56 ± 0.14 ^a^	2.12 ± 0.26 ^b^	1.76 ± 0.08 ^bc^	1.78 ± 0.07 ^bc^	0.90 ± 0.05 ^d^	1.17 ± 0.15 ^c^	**
Vanillic acid	3.32 ± 0.15 ^a^	1.03 ± 0.05 ^b^	0.92 ± 0.24 ^b^	0.87 ± 0.04 ^b^	0.00 ^c^	0.00 ^c^	**
Homovanillic acid	3.22 ± 0.06 ^a^	2.38 ± 0.03 ^b^	2.22 ± 0.22 ^b^	2.19 ± 0.09 ^b^	0.00 ^c^	0.00 ^c^	**
Vanillin	2.77 ± 0.08 ^a^	2.46 ± 0.02 ^b^	2.20 ± 0.05 ^b^	2.31 ± 0.16 ^b^	0.39 ± 0.13 ^c^	1.07 ± 0.07 ^bc^	*
Chlorogenic acid	3.66 ± 0.21 ^a^	2.83 ± 0.12 ^b^	2.38 ± 0.09 ^b^	2.66 ± 0.14 ^b^	2.55 ± 0.27 ^b^	2.25 ± 0.23 ^b^	**
Quercetin 3,4′-Diglucoside	4.45 ± 0.19 ^a^	3.51 ± 0.03 ^c^	3.25 ± 0.20 ^c^	3.85 ± 0.11 ^b^	0.00 ^d^	0.00 ^d^	**
*p*-Coumaric acid	2.32 ± 0.01 ^a^	1.05 ± 0.05 ^bc^	0.93 ± 0.16 ^c^	1.02 ± 0.03 ^b^	0.00 ^d^	0.00 ^d^	**
Ferulic acid	1.88 ± 0.04 ^a^	1.56 ± 0.13 ^b^	1.44 ± 0.06 ^b^	1.36 ± 0.18 ^b^	0.83 ± 0.03 ^c^	1.00 ± 0.07 ^bc^	**
Rutin	5.23 ± 0.22 ^a^	4.99 ± 0.46 ^b^	4.00 ± 0.10 ^e^	4.45 ± 0.53 ^c^	4.25 ± 0.57 ^d^	3.83 ± 0.11 ^f^	**
Luteolin 7-*O*-Glucoside	3.78 ± 0.10 ^c^	2.95 ± 0.09 ^d^	2.62 ± 0.04 ^d^	2.60 ± 0.13 ^d^	4.86 ± 0.64 ^b^	6.75 ± 0.50 ^a^	**
Oleuropein	0.08 ± 0.00 ^b^	0.03 ± 0.00 ^c^	0.03 ± 0.00 ^c^	0.03 ± 0.00 ^c^	0.08 ± 0.00 ^b^	0.17 ± 0.00 ^a^	**
Cinnamic acid	0.65 ± 0.05 ^a^	0.38 ± 0.04 ^b^	0.00 ^c^	0.00 ^c^	0.00 ^c^	0.00 ^c^	**
Pinoresinol	51.34 ± 1.51 ^a^	47.66 ± 1.24 ^b^	46.38 ± 1.95 ^b^	46.51 ± 1.78 ^b^	52.29 ± 0.70 ^a^	47.69 ± 0.76 ^b^	**
Quercetin	2.32 ± 0.14 ^b^	1.57 ± 0.02 ^b^	1.68 ± 0.08 ^b^	3.22 ± 0.11 ^a^	2.46 ± 0.07 ^a^	3.35 ± 0.11 ^a^	**
Apigenin	55.56 ± 2.47 ^b^	51.21 ± 2.07 ^c^	42.96 ± 1.64 ^e^	35.19 ± 12.54 ^f^	58.43 ± 5.17 ^a^	44.63 ± 2.50 ^d^	**
Isoramnetin 3-*O*-Glucoside	0.83 ± 0.09 ^a^	0.47 ± 0.05 ^d^	0.62 ± 0.14 ^bc^	0.68 ± 0.00 ^b^	0.59 ± 0.08 ^cd^	0.55 ± 0.03 ^d^	**
Apigenin 7-O-Glucoside	1.34 ± 0.07 ^a^	1.01 ± 0.06 ^c^	1.21 ± 0.11 ^b^	0.95 ± 0.18 ^c^	0.85 ± 0.05 ^d^	0.13 ± 0.01 ^e^	**
Kaempferol	5.08 ± 0.24 ^a^	4.09 ± 0.04 ^b^	3.57 ± 0.00 ^c^	3.49 ± 0.10 ^c^	2.86 ± 0.05 ^d^	2.81 ± 0.06 ^d^	**
Isoramnetin	2.55 ± 0.11 ^c^	1.02 ± 0.02 ^d^	1.13 ± 0.09 ^d^	2.66 ± 0.14 ^c^	6.88 ± 0.81 ^b^	16.00 ± 3.24 ^a^	**
6-Gingerol	22.56 ± 0.13 ^a^	21.00 ± 0.27 ^b^	19.97 ± 0.76 ^b^	20.79 ± 0.83 ^b^	18.40 ± 0.18 ^c^	23.31 ± 0.26 ^a^	**
6-Shogaol	0.43 ± 0.06 ^a^	0.26 ± 0.15 ^c^	0.21 ± 0.09 ^c^	0.34 ± 0.00 ^ab^	0.12 ± 0.05 ^d^	0.27 ± 0.02 ^bc^	**

Data are expressed as means ± S.D. (*n* = 3). Differences within and between groups were evaluated by one-way ANOVA followed by Tukey’s test: ** *p* < 0.01. Results followed by different letters in a same line are highly significantly different at * *p* ≤ 0.05; ** *p* ≤ 0.01; ns *p* > 0.05 not significant.

**Table 7 foods-12-03822-t007:** Single phenolic compound in GM (ginger olive oil obtained by malaxation) by UHPLC. Values are expressed as mg kg^−1^.

Compounds	T0	T15	T30	T60	T180	T360	Sign
Hydroxytyrosol	26.90 ± 0.90 ^a^	7.21 ± 0.19 ^b^	6.33 ± 0.86 ^b^	8.37 ± 0.93 ^b^	6.59 ± 0.09 ^b^	9.57 ± 0.03 ^b^	**
Tyrosol	20.66 ± 0.16 ^c^	41.07 ± 2.13 ^b^	39.13 ± 3.48 ^b^	41.45 ± 2.65 ^b^	17.56 ± 2.31 ^c^	61.19 ± 4.63 ^a^	**
3,4-Dihydroxybenzoic acid	0.28 ± 0.03 ^a^	0.00 ^b^	0.00 ^b^	0.00 ^b^	0.00 ^b^	0.00 ^b^	**
4-Hydroxyphenyl acetate	3.33 ± 0.10 ^a^	3.29 ± 0.31 ^a^	3.04 ± 0.23 ^a^	3.49 ± 0.47 ^a^	1.00 ± 0.10 ^c^	2.50 ± 0.28 ^b^	**
Caffeic acid	2.21 ± 0.16 ^a^	0.00 ^b^	0.00 ^b^	0.00 ^b^	0.00 ^b^	0.00 ^b^	**
Vanillic acid	6.34 ± 0.34 ^a^	2.56 ± 0.29 ^b^	1.32 ± 0.18 ^c^	0.57 ± 0.02 ^c^	0.66 ± 0.10 ^c^	0.71 ± 0.08 ^c^	**
Homovanillic acid	3.33 ± 0.41 ^a^	2.06 ± 0.09 ^b^	2.36 ± 0.10 ^b^	3.37 ± 0.10 ^a^	0.00 ^c^	0.00 ^c^	**
Vanillin	1.59 ± 0.12 b^c^	3.18 ± 0.04 ^a^	1.96 ± 0.36 ^ab^	2.11 ± 0.10 ^ab^	0.00 ^d^	0.00 ^d^	**
Chlorogenic acid	10.03 ± 0.90 ^a^	10.06 ± 0.74 ^a^	10.73 ± 0.74 ^a^	10.96 ± 0.30 ^a^	4.04 ± 0.00 ^b^	3.16 ± 0.32 ^b^	**
Quercetin 3,4′-Diglucoside	1.93 ± 0.10 ^b^	1.36 ± 0.06 ^b^	4.21 ± 0.50 ^a^	4.61 ± 0.43 ^a^	1.00 ± 0.24 ^b^	0.95 ± 0.04 ^b^	**
*p*-Coumaric acid	0.21 ± 0.03 ^c^	0.39 ± 0.02 ^b^	0.46 ± 0.00 ^a^	0.38 ± 0.08 ^b^	0.00 ^d^	0.00 ^d^	**
Ferulic Acid	0.63 ± 0.04 ^b^	0.39 ± 0.03 ^d^	0.63 ± 0.00 ^b^	0.71 ± 0.10 ^a^	0.52 ± 0.08 ^c^	0.00 ^d^	**
Rutin	1.12 ± 0.12 ^bc^	1.57 ± 0.09 ^b^	2.18 ± 0.21 ^a^	2.06 ± 0.30 ^a^	0.84 ± 0.08 ^c^	0.00 ^d^	**
*o*-Coumaric acid	0.46 ± 0.06 ^c^	0.73 ± 0.04 ^abc^	0.59 ± 0.07 ^bc^	0.62 ± 0.04 ^c^	0.08 ± 0.01 ^a^	0.80 ± 0.04 ^ab^	**
Luteolin 7-*O*-Glucoside	0.21 ± 0.01 ^e^	5.17 ± 0.38 ^b^	6.04 ± 0.00 ^a^	2.83 ± 0.03 ^d^	3.09 ± 0.04 ^c^	0.00 ^f^	**
Oleuropein	0.13 ± 0.02 ^a^	0.09 ± 0.00 ^b^	0.06 ± 0.00 ^c^	0.06 ± 0.00 ^c^	0.04 ± 0.00 ^c^	0.00 ^d^	**
Cinnamic acid	1.07 ± 0.07 ^a^	0.74 ± 0.04 ^b^	0.01 ± 00 ^c^	0.00 ^d^	0.00 ^d^	0.00 ^d^	**
Pinoresinol	9.22 ± 0.31 ^c^	28.54 ± 2.86 ^b^	25.40 ± 1.59 ^b^	26.34 ± 1.17 ^b^	24.41 ± 0.54 ^b^	71.34 ± 5.35 ^a^	**
Luteolin	2.18 ± 0.25 ^a^	0.16 ± 0.05 ^b^	0.11 ± 0.03 ^b^	0.00 ^c^	0.00 ^c^	0.00 ^c^	**
Quercetin	1.59 ± 0.13 ^b^	2.70 ± 0.21 ^b^	0.73 ± 0.01 ^c^	0.74 ± 0.00 ^c^	2.38 ± 0.41 ^b^	7.41 ± 0.30 ^a^	**
Apigenin	35.06 ± 0.77 ^a^	4.69 ± 0.44 ^c^	5.10 ± 2.03 ^c^	6.60 ± 0.67 ^c^	8.77 ± 0.91 ^c^	17.12 ± 1.97 ^b^	**
Isoramnetin 3-*O*-Glucoside	0.86 ± 0.05 ^d^	2.53 ± 0.18 ^c^	2.12 ± 0.51 ^c^	2.66 ± 0.32 ^c^	4.10 ± 1.27 ^a^	3.03 ± 0.73 ^b^	**
Naringenin	3.19 ± 0.13 ^a^	1.07 ± 0.05 ^c^	0.54 ± 0.19 ^c^	1.77 ± 0.74 ^b^	0.00 ^d^	0.00 ^d^	**
Kaempferol	3.09 ± 0.08 ^d^	9.26 ± 0.23 ^c^	11.06 ± 0.51 ^b^	11.89 ± 0.40 ^b^	9.00 ± 2.20 ^c^	22.51 ± 0.24 ^a^	**
Isoramnetin	3.04 ± 0.06 ^c^	6.45 ± 0.22 ^b^	9.49 ± 0.10 ^a^	9.97 ± 0.44 ^a^	9.07 ± 0.57 ^a^	0.22 ± 0.03 ^d^	**
6-Gingerol	5.07 ± 6.03 ^d^	52.93 ± 3.51 ^bc^	55.35 ± 3.89 ^b^	59.14 ± 3.72 ^b^	48.41 ± 0.75 ^c^	128.98 ± 1.78 ^a^	**
6-Shogaol	1.57 ± 0.32 ^c^	2.27 ± 0.24 ^b^	2.87 ± 0.21 ^a^	3.13 ± 0.44 ^a^	1.76 ± 0.03 ^c^	1.74 ± 0.04 ^c^	**
Apigenin 7-*O*-Glucoside	0.80 ± 0.14 ^b^	0.95 ± 0.07 ^a^	0.07 ± 0.04 ^d^	0.13 ± 0.03 ^c^	0.00 ^e^	0.00 ^e^	**

Data are expressed as means ± S.D. (*n* = 3). Differences within and between groups were evaluated by one-way ANOVA followed by Tukey’s test: ** *p* < 0.01. Results followed by different letters in a same line are highly significantly different at * *p* ≤ 0.05; ** *p* ≤ 0.01; ns *p* > 0.05 not significant.

**Table 8 foods-12-03822-t008:** Radical scavenging activity of EVOO and FVOOs against DPPH and ABTS assays during storage. Values are expressed as IC_50_ (μg mL^−1^).

	T0	T15	T30	T60	T180	T360	Sign
DPPH							
EVOO	12.33 ± 3.45 ^bC^	14.09 ± 3.21 ^bC^	15.72 ± 2.87 ^bBC^	20.77 ± 2.82 ^bBC^	19.61 ± 3.09 ^bB^	29.54 ± 3.77 ^bA^	**
GI	17.42 ± 2.27 ^bB^	19.33 ± 2.81 ^abB^	21.05 ± 2.76 ^bB^	37.23 ± 2.08 ^aA^	39.67 ± 2.20 ^aA^	44.21 ± 2.36 ^aA^	**
GM	17.42 ± 2.27 ^aC^	18.8 ±2.32 ^aC^	19.48 ± 2.76 ^abC^	22.22 ± 2.08 ^bCB^	27.63 ± 2.89 ^bB^	35.35 ± 2.94 ^bA^	**
Sign	**	*	*	**	**	**	
ABTS							
EVOO	3.43 ± 0.25 ^bB^	4.98 ± 0.77 ^aB^	5.16 ± 0.93 ^bB^	7.39 ± 0.91 ^B^	11.43 ± 0.86 ^bB^	15.21 ± 1.19 ^bA^	**
GI	4.75 ± 0.24 ^bC^	4.89 ± 0.45 ^aC^	5.86 ± 0.27 ^abC^	7.51 ± 0.14 ^B^	8.32 ± 0.67 ^bB^	11.31 ± 1.09 ^bA^	**
GM	4.75 ± 0.24 ^aC^	5.61 ± 0.28 ^aC^	6.07 ± 0.34 ^aC^	7.62 ± 0.57 ^C^	18.28 ± 1.13 ^aB^	26.31 ± 1.47 ^aA^	**
Sign	*	*	*	ns	**	**	

Data are expressed as means ± S.D. (*n* = 3). EVOO: control; GI: ginger olive oil obtained by infusion; GM: ginger olive oil obtained by malaxation. Ascorbic acid was used as positive control in both DPPH and ABTS tests (IC_50_ values of 5.03 ± 0.82 and 1.78 ± 0.07 μg mL^−1^, respectively). Results followed by different capital letters in the same row show the differences in one sample during storage. The different lowercase letters in the same column show the differences among the samples at the same time. Differences within and between groups were evaluated by one-way ANOVA followed by Tukey’s test: ** *p* < 0.01. Results followed by different letters are highly significantly different at * *p* ≤ 0.05; ** *p* ≤ 0.01; ns *p* > 0.05 not significant.

**Table 9 foods-12-03822-t009:** Evaluation of EVOO and FVOO protection from lipid peroxidation evaluated by β-carotene bleaching test. Values are expressed as IC_50_ (μg mL^−1^).

	T0	T15	T30	T60	T180	T360	Sign
EVOO	48.72 ± 3.45 ^aD^	52.21 ± 3.89 ^aD^	59.8 3 ± 4.40 ^aC^	77.05 ± 4.42 ^aB^	>100 ^aA^	>100 ^aA^	**
GI	18.68 ± 2.59 ^aD^	19.98 ± 2.71 ^bD^	23.41 ± 2.19 ^bCD^	27.72 ± 2.08 ^bBC^	32.09 ± 2.11 ^cB^	46.10 ± 2.80 ^cA^	**
GM	18.68 ± 2.59 ^bE^	20.09 ± 2.69 ^bE^	27.92 ± 2.75 ^bD^	33.12 ± 3.08 ^bC^	50.96 ± 3.88 ^bB^	77.67 ± 4.09 ^bA^	**
Sign	**	**	**	**	**	**	

Data are expressed as means ± S.D. (*n* = 3). EVOO: control; GI: ginger olive oil obtained by infusion; GM: ginger olive oil obtained by malaxation. Propyl gallate (IC_50_ values of 1.02 ± 0.01 μg mL^−1^) was used as positive control. Results followed by different capital letters in the same row show the differences in one sample during storage. The different lowercase letters in the same column show the differences among the samples at the same time. Differences within and between groups were evaluated by one-way ANOVA followed by Tukey’s test: ** *p* < 0.01. Results followed by different letters are highly significantly different at * *p* ≤ 0.05; ** *p* ≤ 0.01; ns *p* > 0.05 not significant.

**Table 10 foods-12-03822-t010:** Carbohydrate hydrolysing enzymes (α-amylase and α-glucosidase) inhibitory activity. Values are expressed as IC_50_ (μg mL^−1^).

	T0	T15	T30	T60	T180	T360	Sign
α-amylase							
EVOO	269.02 ± 3.77 ^aE^	275.21 ± 3.85 ^aD^	303.38 ± 3.92 ^aB^	345.31 ± 4.05 ^aA^	240.29 ± 3.87 ^bF^	289.32 ± 4.90 ^cC^	**
GI	126.95 ± 3.56 ^aD^	131.23 ± 3.87 ^bD^	170.47 ± 3.44 ^bC^	256.93 ± 3.35 ^bB^	263.22 ± 3.77 ^aB^	305.11 ± 4.09 ^bA^	**
GM	126.93 ± 3.56 ^bE^	131.09 ± 3.68 ^bE^	155.89 ± 3.44 ^cD^	175.06 ± 3.35 ^cC^	220.17 ± 2.22 ^cB^	328.10 ± 3.55 ^aA^	**
Sign	**	**	**	**	**	**	
α-glucosidase							
EVOO	137.34 ± 3.73 ^bF^	145.18 ± 3.79 ^bE^	198.81 ± 3.82 ^D^	337.56 ± 3.90 ^aC^	587.49 ± 3.56 ^aB^	778.23 ± 4.67 ^aA^	**
GI	181.67 ± 3.45 ^bD^	184.12 ± 3.87 ^aDC^	193.46 ± 3.09 ^C^	208.11 ± 3.01 ^bB^	219.36 ± 3.20 ^cB^	269.71 ± 3.85 ^cA^	**
GM	181.67 ± 3.45 ^aE^	185.90 ± 3.67 ^aDE^	196.74 ± 3.89 ^D^	210.71 ± 4.01 ^bC^	235.54 ± 4.89 ^bB^	407.89 ± 5.08 ^bA^	**
Sign	**	**	ns	**	**	**	

Data are expressed as means ± S.D. (*n* = 3). EVOO: control; GI: ginger olive oil obtained by infusion; GM: ginger olive oil obtained by malaxation. Acarbose was used as positive control in both tests with IC_50_ values of 50.18 ± 1.32 and 35.57 ± 0.99 μg mL^−1^ for α-amylase and α-glucosidase. Results followed by different capital letters in the same row show the differences in one sample during storage. The different lowercase letters in the same column show the differences among the samples at the same time. Differences within and between groups were evaluated by one-way ANOVA followed by Tukey’s test: ** *p* < 0.01. Results followed by different letters are highly significantly different at * *p* ≤ 0.05; ** *p* ≤ 0.01; ns *p* > 0.05 not significant.

**Table 11 foods-12-03822-t011:** Lipase assay during storage. Values are expressed as IC_50_ (μg mL^−1^).

	T0	T15	T30	T60	T180	T360	Sign
EVOO	143.46 ± 4.85 ^aF^	155.52 ± 4.87 ^aE^	173.43 ± 4.91 ^aD^	206.54 ± 5.01 ^aC^	253.81 ± 4.81 ^aB^	312.97 ± 5.44 ^aA^	**
GI	63.45 ± 4.09 ^aD^	65.07 ± 4.26 ^bD^	107.93 ± 4.22 ^bC^	167.82 ± 4.02 ^cB^	169.56 ± 4.14 ^bB^	195.96 ± 4.77 ^bA^	**
GM	63.45 ± 1.09 ^bE^	65.48 ± 1.15 ^bE^	79.36 ± 1.22 ^cD^	91.94 ± 1.02 ^bC^	110.95 ± 2.46 ^cB^	309.21 ± 2.87 ^aA^	**
Sign	**	**	**	**	**	**	

Data are expressed as means ± S.D. (*n* = 3). EVOO: control; GI: ginger olive oil obtained by infusion; GM: ginger olive oil obtained by malaxation. Orlistat was used as positive control (IC_50_ value of 37.44 ± 1.08 μg mL^−1^). Results followed by different capital letters in the same row show the differences in one sample during storage. The different lowercase letters in the same column show the differences among the samples at the same time. Differences within and between groups were evaluated by one-way ANOVA followed by Tukey’s test: ** *p* < 0.01. Results followed by different letters are highly significantly different at * *p* ≤ 0.05; ** *p* ≤ 0.01; ns *p* > 0.05 not significant.

## Data Availability

The data presented in this study are available on request from the corresponding author.

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
