# Peer review of "Evaluation of Quality Parameters and Functional Activity of Ottobratica Extra Virgin Olive Oil Enriched with *Zingiber officinale* (Ginger) by Two Different Enrichment Processes during One-Year Storage"

_foods, 2023, doi:10.3390/foods12203822_

Round 1
Reviewer 1 Report
Comments and Suggestions for Authors
The purpose of this study is to evaluate the effect of the addition of ginger root powder on the quality parameters and nutritional value of oil that has been stored for 1 year. The research topic is of practical importance, and the amount of research described in the article is very large. However, the article needs some additions, a list of which is presented below.
The introduction presents recent research on EVOO, but there is no information on whether ginger was previously added during production, or to the finished product. If such studies were previously conducted, they should be mentioned. If, on the other hand, no such research has been conducted before, then the innovative nature of such research should be emphasized.
The description of the research material lacks detailed information on ginger root powder, and the description of this material is very crucial to the research. Information such as country of origin, year of production, dry content of substances is missing. All the information provided by the manufacturer would be helpful in assessing the quality of this powder.
The FVOOs were stored at room temperature in 100 mL green glass bottles. Since this is a storage experiment, information on the degree to which the bottles were filled with oil is missing. What was the headspace in the bottles?
I recommend replacing rpm with g, when centrifuging (L113)
L150, L176 please remove the dot
In the description of the method for the determination of phenolic profile determined on UHPLC, there is no information on what wavelength was used, the temperature of the measurement and the phase flow. Also missing is information on what standards were used in the determination.
I recommend that a description of the abbreviations GM and GI be added to the description of the figures. The development of the abbreviations is in the text of the manuscript, but the graphs would be more readable.
Please check the statistical designations in figure 7. EVOO - **7
Why in the description of the methodology is described in the case of ginger root powder is TPC and TCC expressed in mg/g DW. In contrast, the results in Table 2 are given in ginger extract and expressed in mg/kg. So all the results are converted to the extract obtained? This needs a clear explanation in the description of the analytical methods.
The phenolic profile in the oils tested is very interesting. I propose to include in the additional files chromatograms showing the phenolic profile of ginger, EVOO, GI, and GM.
I recommend that the description of the sensory analysis performed be moved from the results description section to the description of the analytical methods.
I recommend giving the number of the ethical approval reference, when describing the sensory analysis performed.
Author Response
Dear editor and reviewers,
We thank you for your critical comments and thoughtful suggestions. Based on these comments and suggestions, we have made careful modifications on the original manuscript and the revised version of the manuscript has been resubmitted to your journal. All changes made to the text are in red color. We hope the revised manuscript will meet the journal’s standard. Below you will find our point-by-point responses to the comments/questions, we sincerely hope to get your approval. If you have any comments/questions, please contact us, we’ll make serious changes.
Q: The introduction presents recent research on EVOO, but there is no information on whether ginger was previously added during production, or to the finished product. If such studies were previously conducted, they should be mentioned. If, on the other hand, no such research has been conducted before, then the innovative nature of such research should be emphasized.
A: Corrected in the text.
Q: The description of the research material lacks detailed information on ginger root powder, and the description of this material is very crucial to the research. Information such as country of origin, year of production, dry content of substances is missing. All the information provided by the manufacturer would be helpful in assessing the quality of this powder.
A: Corrected in the text.
Q: The FVOOs were stored at room temperature in 100 mL green glass bottles. Since this is a storage experiment, information on the degree to which the bottles were filled with oil is missing. What was the headspace in the bottles?
A: The headspace was 1 cm3.
Q: I recommend replacing rpm with g, when centrifuging (L113)
A: Corrected in the text.
Q: L150, L176 please remove the dot
A: Corrected in the text.
Q: In the description of the method for the determination of phenolic profile determined on UHPLC, there is no information on what wavelength was used, the temperature of the measurement and the phase flow. Also missing is information on what standards were used in the determination.
A: Corrected in the text.
Q: I recommend that a description of the abbreviations GM and GI be added to the description of the figures. The development of the abbreviations is in the text of the manuscript, but the graphs would be more readable.
A: Corrected in the text.
Q: Please check the statistical designations in figure 7. EVOO - **7
A: Corrected in the text.
Q: Why in the description of the methodology is described in the case of ginger root powder is TPC and TCC expressed in mg/g DW. In contrast, the results in Table 2 are given in ginger extract and expressed in mg/kg. So all the results are converted to the extract obtained? This needs a clear explanation in the description of the analytical methods.
A: There was an error in the table caption. It has now been corrected.
Q: The phenolic profile in the oils tested is very interesting. I propose to include in the additional files chromatograms showing the phenolic profile of ginger, EVOO, GI, and GM.
A: The requested chromatograms were attached in supplementary material.
Q: I recommend that the description of the sensory analysis performed be moved from the results description section to the description of the analytical methods.
A: Corrected in the text.
Q: I recommend giving the number of the ethical approval reference, when describing the sensory analysis performed.
A: The sensory analysis was done in accordance with the current legislation and according to the internal regulations of the department. All the panelists were previously informed on the ingredients they tasted.

Reviewer 2 Report
Comments and Suggestions for Authors
In this paper, Custureri et al. investigated the effect of enriching a variety of EVOO with ginger by two different enrichment processes on quality characteristics and in vitro bioactivity.
Major Comments :
1) Line 74: Since several authors have confirmed that adding plant material to the malaxer is more efficient in terms of quality than adding it by infusion in the olive oil, what makes this study unique? Please add relevant information in the Introduction section.
2) Please explain why you chose to add ginger root powder in different concentration in the two enrichment processes, i.e. 1% in the malaxer and 2% by infusion.
3) Since there is a large amount of results, maybe consider separating Results from Discussion so as not to confuse the reader of the manuscript.
Minor comments
1) Introduction : Please add sufficient background and relevant references regarding the in vitro bioactivity.
2) Figures 3,4,6,7,8 : there is no description in y-axis.
Comments on the Quality of English Language
English language is generally fine. Only minor editing is required.
Author Response
Dear editor and reviewers,
We thank you for your critical comments and thoughtful suggestions. Based on these comments and suggestions, we have made careful modifications on the original manuscript and the revised version of the manuscript has been resubmitted to your journal. All changes made to the text are in red color. We hope the revised manuscript will meet the journal’s standard. Below you will find our point-by-point responses to the comments/questions, we sincerely hope to get your approval. If you have any comments/questions, please contact us, we’ll make serious changes.
Q: Line 74: Since several authors have confirmed that adding plant material to the malaxer is more efficient in terms of quality than adding it by infusion in the olive oil, what makes this study unique? Please add relevant information in the Introduction section.
A: Corrected in the text.
Q: Please explain why you chose to add ginger root powder in different concentration in the two enrichment processes, i.e. 1% in the malaxer and 2% by infusion.
A: After different preliminary laboratory tests, the best results were obtained with these enrichment percentages (1 and 2% by malaxation and by infusion, respectively).
Q: Since there is a large amount of results, maybe consider separating Results from Discussion so as not to confuse the reader of the manuscript.
A: Thanks for the suggestion, but we think separating results and discussion creates more confusion.
Q: Introduction: Please add sufficient background and relevant references regarding the in vitro bioactivity.
A: Corrected in the text.
Q: Figures 3,4,6,7,8: there is no description in y-axis.
A: Corrected in the text.

Reviewer 3 Report
Comments and Suggestions for Authors
In my opinion the article needs some improvements.
Some examples of improvements are presented below:
- Row 19: Please explain the acronym EVOO. Acronyms should be defined the first time they appear in each of three sections: the abstract; the main text; the first figure or table. Then only the acronym is used in the text. Please look for this aspect in the whole article.
- Row 27: Please replace T60 with after 60 days of storage.
- Row 34: Please explain the acronyms EVOO, MUFA and PUFA. Acronyms should be defined the first time they appear in each of three sections: the abstract; the main text; the first figure or table. Then only the acronym is used in the text. Please look for this aspect in the whole article.
- Row 71: Please add the reference [12].
- Row 73: Is the comma after references 3,4 necessary?
- I suggest you pay attention to the spaces between words in the hole articles (e.g. Rows 150, 176). Is the point before tocopherol / carotene necessary?
- Row 152: The symbol for microns is “µm”.
- I suggest redraw the figures (example: color graphs, give up the gridlines, the names of the axis, axis formatting, the major units on the vertical axis: maybe it would be good if they were big).
- Row 410: C. annuum in italic, please.
- Please do not use bibliographic references in the Conclusions section.
- Acronyms should be defined the first time they appear in each of three sections: the abstract; the main text; the first figure or table. Abbreviations section is necessary?
- Please add more recent references.
Author Response
Dear editor and reviewers,
We thank you for your critical comments and thoughtful suggestions. Based on these comments and suggestions, we have made careful modifications on the original manuscript and the revised version of the manuscript has been resubmitted to your journal. All changes made to the text are in red color. We hope the revised manuscript will meet the journal’s standard. Below you will find our point-by-point responses to the comments/questions, we sincerely hope to get your approval. If you have any comments/questions, please contact us, we’ll make serious changes.
Q: Row 19: Please explain the acronym EVOO. Acronyms should be defined the first time they appear in each of three sections: the abstract; the main text; the first figure or table. Then only the acronym is used in the text. Please look for this aspect in the whole article.
A: Corrected in the text.
Q: Row 27: Please replace T60 with after 60 days of storage.
A: Corrected in the text.
Q: Row 34: Please explain the acronyms EVOO, MUFA and PUFA. Acronyms should be defined the first time they appear in each of three sections: the abstract; the main text; the first figure or table. Then only the acronym is used in the text. Please look for this aspect in the whole article.
A: Corrected in the text.
Q: Row 71: Please add the reference [12].
A: Corrected in the text.
Q: Row 73: Is the comma after references 3,4 necessary?
A: Corrected in the text.
Q: I suggest you pay attention to the spaces between words in the hole articles (e.g. Rows 150, 176). Is the point before tocopherol / carotene necessary?
A: Corrected in the text.
Q: Row 152: The symbol for microns is “µm”.
A: Corrected in the text.
Q: I suggest redraw the figures (example: color graphs, give up the gridlines, the names of the axis, axis formatting, the major units on the vertical axis: maybe it would be good if they were big).
A: Corrected in the text.
Q: Row 410: C. annuum in italic, please.
A: Corrected in the text.
Q: Please do not use bibliographic references in the Conclusions section.
A: Corrected in the text.
Q: Acronyms should be defined the first time they appear in each of three sections: the abstract; the main text; the first figure or table. Abbreviations section is necessary?
A: Corrected in the text.
Q: Please add more recent references.
A: Corrected in the text.

Round 2
Reviewer 1 Report
Comments and Suggestions for Authors
All my comments were taken into account in the revised manuscript. However, I believe that additional files need to be changed. Including only chromatographs with descriptions not in English does not add anything valuable to the article. Chromatograms should be deleted or appropriate signatures and names of identified peaks should be entered.
Author Response
You thank you for your comments. Based on these suggestions, we have made careful modifications.
Please see the attachment,
